# BIT-PRUNING:
# A SPARSE MULTIPLICATION-LESS DOT-PRODUCT

**Yusuke Sekikawa, Shingo Yashima**
DENSO IT LABORATORY, INC., Tokyo, Japan
{sekikawa.yusuke, yashima.shingo}@core.d-itlab.co.jp

## ABSTRACT

Dot-product is a central building block in neural networks. However, multiplication (`mult`) in dot-product consumes intensive energy and space costs that challenge deployment on resource-constrained edge devices. In this study, we realize energy-efficient neural networks by exploiting a `mult`-less, sparse dot-product. We first reformulate a dot-product between an integer weight and activation into an equivalent operation comprised of additions followed by bit-shifts (`add-shift-add`). In this formulation, the number of `add` operations equals the number of bits of the integer weight in binary format. Leveraging this observation, we propose Bit-Pruning, which removes unnecessary bits in each weight value during training to reduce the energy consumption of `add-shift-add`. Bit-Pruning can be seen as soft Weight-Pruning as it prunes bits, not the whole weight element. In extensive experiments, we demonstrate that sparse `mult`-less networks trained with Bit-Pruning show a better accuracy-energy trade-off than sparse `mult` networks trained with Weight-Pruning. (Code is available at https://github.com/DensoITLab/bitprune)

## 1 INTRODUCTION

Modern deep neural networks (DNNs) contain numerous dot-products between input features and weight matrices. However, it is well known that multiplication (`mult`) in dot-product consumes intensive energy and space costs, challenging DNNs' deployment on resource-constrained edge devices. This drives several attempts for efficient DNNs by reducing the energy of `mult`.

*Quantization* (Yin et al., 2019; Esser et al., 2020; Li & Baillieul, 2004) discretizes the weight and/or activation into a low-bit representation; low-precision `mult` requires less energy than the high-precision counterpart. *Power of two networks* (Li et al., 2019b; Miyashita et al., 2016; Elhoushi et al., 2021) restricts the `mult` operation to the power of two (PoT); PoT `mult` can be realized by a bit-shift (`shift`), which consumes orders of magnitude less energy than `mult`. Although these approaches have successfully reduced energy consumption, they *define* the energy-efficient model structure *before training*. That is, they limit model capacity and impose training with a precision that is difficult to use gradient-based optimization, e.g., approximate gradient with the straight-through estimator (STE) (Yin et al., 2019), making it challenging to achieve good accuracy in the low-bit regime (e.g., 4bit, 2bit, or binary).

Table 1: Energy [*pJ*] and area [μ*m*²] cost on ASIC (45nm technology) and FPGA (ZYNQ-7ZC706). Data adapted from (You et al., 2020; Gholami et al., Horowitz, 2014)

| Operation | Format | ASIC | | FPGA |
| --- | --- | --- | --- | --- |
| | | Energy | Area | Energy |
| `mult` | FIX32 | 3.1 | 3495 | 19.6 |
| | FIX8 | 0.2 | 282 | 0.2 |
| `add` | FIX32 | 0.1 | 137 | 0.1 |
| | FIX8 | 0.03 | 36 | 0.1 |
| `shift` | FIX32 | 0.13 | - | 0.1 |
| | FIX8 | 0.024 | - | 0.025 |

Aside from low-bit approaches, *Weight-Pruning* (Frankle & Carbin, 2018; Chen et al., 2022; Yang et al., 2020; Wortsman et al., 2019) *learns* efficient model structure *during training* by removing unimportant weight from high capacity models. Owing to its data-driven nature, it could remove the unnecessary `mult` without sacrificing accuracy (e.g., 95% reduction in unstructured pruning). Empowered by the recent progress of in/near-memory computing architectures (Gholami et al.),

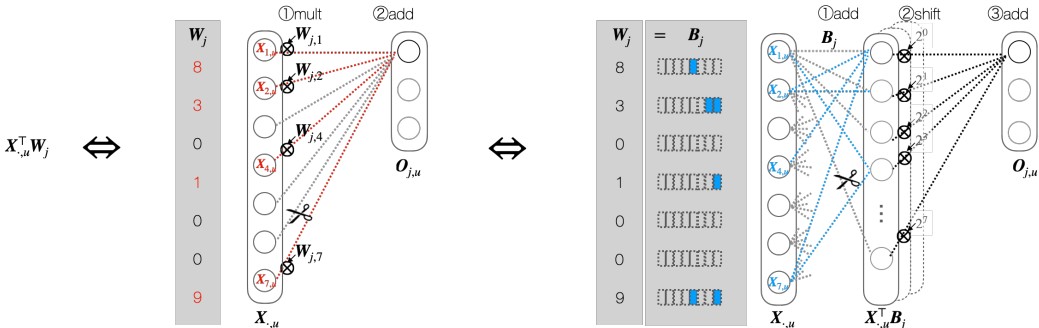

Figure 1: Dot-product realized using sparse `mult-add` (left) and equivalent representation using `add-shift-add`(right). In the `mult-add`, computation is reduced by pruning the entire weight (`mult`). In contrast, in the `add-shift-add` computation is reduced by pruning the bit (`add`) in binary format.

several DNN accelerators designed for the unstructured sparsity have emerged (Han et al., 2016; Bamberg et al., 2023; Dietrich et al., 2021; Zhang et al., 2020), realizing the efficient inference of these sparse networks. Furthermore, optimized implementation for general-purpose CPUs also proved to be a good candidate for unstructured sparsity (Kurtz et al., 2020).

Motivated by the significant energy reduction of sparse `mult` operations by Weight-Pruning and recent advances in the frameworks supporting unstructured sparsity, we envision new frontiers in the accuracy/energy trade-off by realizing *sparse* and *`mult`-less* dot-product for hardware supporting the unstructured sparsity. Noting that the dot-product between integer weights and activation can be decomposed into a PoT basis and binary vector, we first reformulate a dot-product between integer weight and activation (`mult-add`) into an equivalent operation comprised of additions followed by bit-shifts and additions (`add-shift-add`, Figure 1). In this formulation, the number of the first `add` operations equals the `bitcount` of the weight elements in binary format. From this observation, we propose Bit-Pruning, which removes unnecessary `add` (therefore unnecessary bits in weight) during training in a data-driven manner to reduce the energy consumption in the `add-shift-add` operation. Because of the difficulty of optimization in binary format, we optimize parameters on a high-precision differentiable `mult-add` network with bit-sparsity regularization, which promotes weights to be sparse in binary format. After training, the obtained bit-sparse `mult-add` network is converted to an equivalent `add-shift-add` network comprised of sparse `add` for efficient inference on DNN accelerators supporting unstructured sparsity.

As depicted in Figure 1, Bit-Pruning can be seen as a soft and fine-grained Weight-Pruning; it does not necessarily remove the whole weight elements but removes only some unnecessary bits of the weight values. This interpretation raises the following question:

> *Which pruning offers a better accuracy/energy trade-off? `mult` in `mult-add` network (Weight-Pruning) or `add` in `add-shift-add` network (Bit-Pruning)?*

We conducted an extensive evaluation to answer the question (Section 4); the results suggest that pruning the `add` is several times more energy efficient than pruning the `mult`.

***Remark***: We do not argue that the `add-shift-add` representation itself is efficient. In fact, the estimated energy consumption of (dense) `mult-add` and `add-shift-add` without pruning are almost the same, as we see in Section 3.3. Our research interest is to investigate whether the fine-grained pruning by removing bits rather than weights can find a more efficient sparse structure; the `add-shift-add` representation permits this investigation.

## 2 PRELIMINARIES

### 2.1 DOT-PRODUCT WITH MULT-ADD

The dot-product is a fundamental building block of neural networks. In this study, we mainly focus on the dot-product that appears in convolution[1]. Let the weights and activation be quantized to $M$

---

[1]The same discussion can be applied for dot-product in another computation block such as multi-layer perceptron or self/cross attention.

and $N$ bits, respectively, where $M, N$ are sufficiently fine quantization levels such as $M, N = 8, 32$. Convolution with the input $\boldsymbol{I} \in \mathbb{Z}^{C_i \times W \times H}$ and weight $\boldsymbol{W} \in \mathbb{Z}^{C_o \times C_i \times k \times k}$ is equivalent to matrix multiplication of unfolded input $\boldsymbol{X} \in \mathbb{Z}^{C_i kk \times WH}$ and the reshaped weight $\boldsymbol{W} \in \mathbb{Z}^{C_o \times C_i kk}$:

$$\boldsymbol{O} = \boldsymbol{I} * \boldsymbol{W} = \boldsymbol{W}\boldsymbol{X}, \tag{1}$$

For brevity, we consider a single input pixel $u$ and an output channel $j$. Then, the convolution which corresponds to the dot-product of the $u$-th column vector of the unfolded input $\boldsymbol{X}_{\cdot,u}$ and $j$-th row vector of the weight $\boldsymbol{W}_j$ is written as follows:

$$\boldsymbol{O}_{j,u} = \boldsymbol{X}_{\cdot,u}^\top \boldsymbol{W}_j = \sum_{k=1}^{K} \boldsymbol{X}_{k,u} \cdot \boldsymbol{W}_{j,k}, \tag{2}$$

where $K = C_i kk$. Computing (2) using naive dense `mult-add` requires $K$ `mult` followed by $(K-1)$ `add`, which consumes a lot of energy as it contains many `mult` operations (Table 1).

## 2.2 SPARSE MULT-ADD BY WEIGHT-PRUNING

When a DNN processor supports unstructured weight and dynamic activation sparsity (Han et al., 2016; Bamberg et al., 2023), `mult` can be skipped either when the activation or weight is zero. In this case, the number of `mult` operation become $\sigma_{\texttt{mult}} K$ where $\sigma_{\texttt{mult}}$ is the ratio of the nonzero element defined as $\texttt{bitcount}(\boldsymbol{W}_j \neq 0 \mid \boldsymbol{X}_{\cdot,u} \neq 0)/K$. Therefore, the dot-product using sparse `mult-add` consumes less energy than dense `mult-add`.

The sparse `mult-add` dot-product is typically realized by *Weight-Pruning*, which trains a network with the *weight-sparsity regularization* $\mathcal{L}_{wgt}$[2] (Liu et al., 2015; Han et al., 2015):

$$\mathcal{L}(\mathcal{W}) = \mathcal{L}_{task}(\mathcal{W}) + \eta \, \mathcal{L}_{wgt}(\mathcal{W}),$$

$$\mathcal{L}_{wgt}(\mathcal{W}) = \sum_{l=1}^{L} \left| \boldsymbol{X}^{(l)} \odot \boldsymbol{W}^{(l)} \right|_0, \tag{3}$$

where $\mathcal{L}_{task}$ is a loss specific to the given task (e.g., cross-entropy for classification), $\mathcal{W} = \{\boldsymbol{W}^{(l)}\}_{l=1}^{L}$ is a set of weight matrices from all $L$ layers, $\boldsymbol{X}^{(l)}$ is an input of $l$-th layer, and $\eta$ is a hyper-parameter to balance the accuracy and computational energy. In practice, $l_1$-norm or Hoyer(-Square) loss (Yang et al., 2020) is applied to relax $l_0$-norm for efficient training. Note that when the target hardware only supports unstructured weight sparsity (no activation sparsity), the same argument can be applied by replacing $\boldsymbol{X}^{(l)} \odot$ with $\texttt{numel}(\boldsymbol{X}^{(l)})$.

## 3 BIT-PRUNING

For more efficient DNN inference, we propose a sparse and `mult`-less dot-product by *softening* the Weight-Pruning. Our intuition is that better energy-accuracy tradeoff could be achieved by bit-level *fine-grained* pruning instead of the weight-level *course* pruning. We first reformulate a dot-product between integer weight and activation into an equivalent operation comprised of `add-shift-add`, which does not contain `mult` (Section 3.1 which consumes lot of energy, Figure 1). In this formulation, we can see that the number of the first `add` (which directly translates to energy consumption) equals the `bitcount` of the weight element in binary format. Then, we propose *Bit-Pruning*, which removes unnecessary bits of weight elements during training to achieve efficient inference (Section 3.2).

## 3.1 REFORMULATE DOT-PRODUCT WITH ADD-SHIFT-ADD

An $M$-bit weight matrix[3] $\boldsymbol{W} \in \mathbb{Z}^{C_o \times K}$ is represented in binary format with a tensor $\boldsymbol{B} \in \{0,1\}^{C_o \times M \times K}$, where $\boldsymbol{B}_{j,\cdot,k} \in \{0,1\}^M$ is the binary representation of the weight value $\boldsymbol{W}_{j,k} \in \mathbb{Z}$:

$$\boldsymbol{W}_{j,k} = \sum_{m=1}^{M} 2^{m-1} \cdot \boldsymbol{B}_{j,m,k}. \tag{4}$$

---

[2]For $x \in \mathbb{Z}^{I \times J}$ and $y \in \mathbb{Z}^{J \times K}$, $x \odot y \in \mathbb{Z}^{I \times J \times K}$ is defined as $(x \odot y)_{i,j,k} = x_{i,j} \cdot y_{j,k}$.

[3]For simplicity, we assume the weight element $\boldsymbol{W}_{j,k}$ is positive in the following explanation.

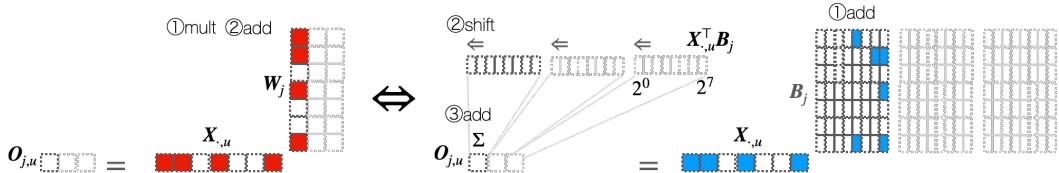

Figure 2: Matrix view of the `mult-add` (left) and `add-shift-add` (right) dot-product.

Then, we rewrite the `mult-add` dot-product (2) into the following `add-shift-add` formulation:

$$\boldsymbol{X}_{\cdot,u}^{\top}\boldsymbol{W}_j = \sum_{k=1}^{K}\left(\boldsymbol{X}_{k,u} \cdot \sum_{m=1}^{M} 2^{m-1} \cdot \boldsymbol{B}_{j,m,k}\right)$$

$$= \sum_{m=1}^{M} 2^{m-1} \cdot \left(\boldsymbol{X}_{\cdot,u}^{\top}\boldsymbol{B}_{j,m}\right). \tag{5}$$

This is computed first by accumulating the input $\boldsymbol{X}_{\cdot,u}$ along the binary vector $\boldsymbol{B}_{j,m}$ using `add`, then multiplying $2^m$ using `shift`, and finally aggregating the results of all $M$ bits by `add`. When the operation is mapped into the hardware supporting the unstructured sparsity, the first `add` using the binary vector $\boldsymbol{B}_{j,m}$ requires `bitcount`($\boldsymbol{B}_{j,m}$) times `add`. The `shift` and last `add` are executed for $M-1$ times (Table 2).

## 3.2 SPARSE ADD-SHIFT-ADD BY BIT-PRUNING

In the `add-shift-add` formulation of (5), the number of the first `add` operations equals the `bitcount` of the binary tensor $\boldsymbol{B}$, which directly translates to energy consumption. Therefore, our goal is to learn the sparse binary tensor of the `add-shift-add` network that also achieves good performance by minimizing the following loss:

$$\mathcal{L}(\mathcal{B}) = \mathcal{L}_{task}(\mathcal{B}) + \eta \ \mathtt{bitcount}(\mathcal{B}), \tag{6}$$

where $\mathcal{B} = \{\boldsymbol{B}^{(l)}\}_{l=1}^{L}$ is a set of binary tensors from all $L$ layers. However, binary parameters prohibit gradient computation, making it difficult to train with standard backpropagation.

Instead of directly optimizing the binary tensor $\boldsymbol{B}$, as done by Yang et al. (2021), we deal with the original weight matrix $\boldsymbol{W}$ while promoting it to be sparse in the binary format $\boldsymbol{B}$. Notice that any dot-product with the integer weight can be precisely expressed as the `add-shift-add` dot-product, and the number of `add` counts for $\boldsymbol{B}$ depends on the corresponding integer weight $\boldsymbol{W}$. For example, when $\boldsymbol{W}_{j,k}$ is zero, then $\boldsymbol{B}_{j,\cdot,k} = [0,0,0,0,0,0,0]^{\top}$, and there is no `add`. When $\boldsymbol{W}_{j,k}$ is PoT (e.g., $2^3$), then $\boldsymbol{B}_{j,\cdot,k} = [0,0,0,1,0,0,0]^{\top}$, and there is only one `add`. In such a way, a number of `add` in `add-shift-add` dot-product depends on the value of the corresponding weight $\boldsymbol{W}_{j,k}$ (Figure 3a), and there is *favorable* value in terms of energy consumption.

We leverage this to propose a framework called Bit-Pruning, which optimizes the network's energy consumption in a data-driven manner. The binary tensor $\boldsymbol{B}$ of the `add-shift-add` network is trained as the high-precision `mult-add` network having weight $\boldsymbol{W}$ (Figure 2), and a sparsification of $\boldsymbol{B}$ is realized by imposing the *bit-sparsity regularization* $\mathcal{L}_{bit}$ on weight $\boldsymbol{W}$:

$$\mathcal{L}(\mathcal{W}) = \mathcal{L}_{task}(\mathcal{W}) + \eta \ \mathcal{L}_{bit}(\mathcal{W}),$$

$$\mathcal{L}_{bit}(\mathcal{W}) = \sum_{l=1}^{L} \left|\boldsymbol{X}^{(l)} \odot (\boldsymbol{W}^{(l)} - \hat{\boldsymbol{W}}^{(l)})\right|_0, \tag{7}$$

where $\mathcal{L}_{bit}$ measures a deviation of the current weight $\boldsymbol{W}$ from bit-sparse *proximal weight* $\hat{\boldsymbol{W}}$, explained as follows. Similar to Weight-Pruning (3), we relax the $l_0$-norm in (7) with $l_1$-norm or Hoyer(-Square) loss in practice.

**Proximal weight** Given the current weight $\boldsymbol{W}$, the proximal weight $\hat{\boldsymbol{W}}$ is computed (element-wise) as a minimizer of the sum of *weight proximity cost* $\mathcal{C}_{mv}$ and `add` *count cost* $\mathcal{C}_{add}$:

$$\hat{\boldsymbol{W}}_{j,k} = \arg\min_{w}\left(\mathcal{C}_{mv}(w, \boldsymbol{W}_{j,k}) + \lambda^{(l)}\mathcal{C}_{add}(w)\right), \tag{8}$$

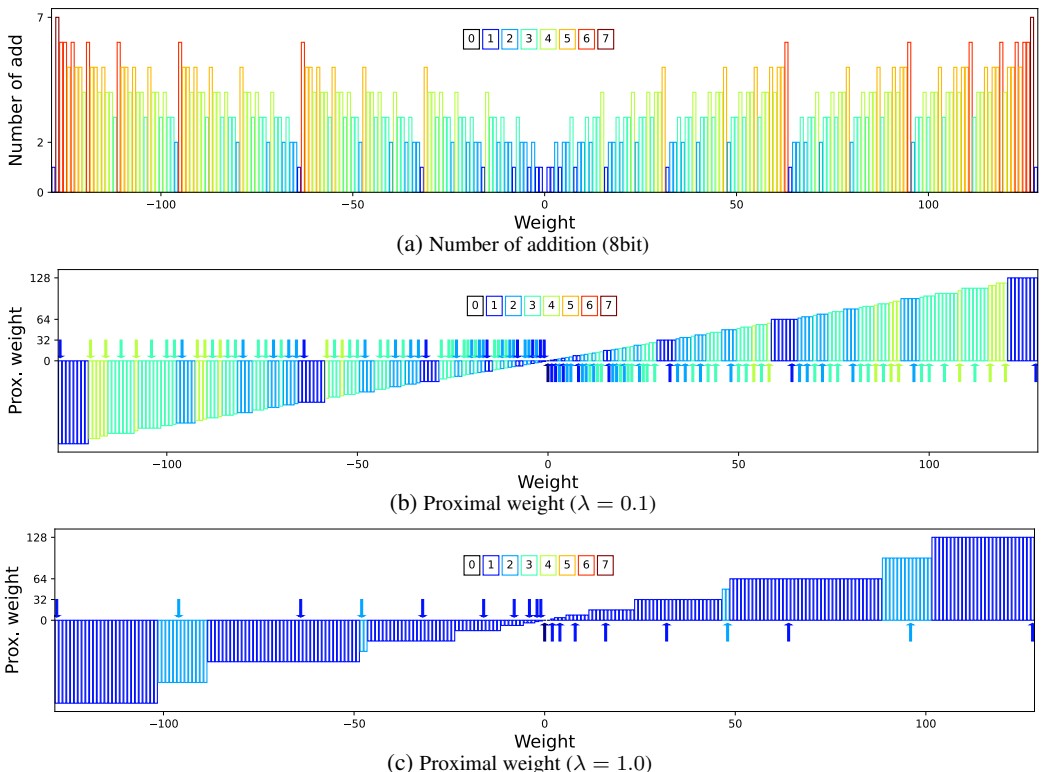

Figure 3: (a) Number of `add` for each weight (`add` count cost $\mathcal{C}_{add}$). (b)-(c) Proximal weight $\hat{\boldsymbol{W}}$ (arrow), which is a solution of (8) (number of `add` for `add-shift-add` dot-product is encoded as color). Refer to Appendix C for the visualization of corresponding loss landscape ($\mathcal{C}_{mv}(w, \hat{w}) + \mathcal{C}_{add}(\hat{w})$).

where $\mathcal{C}_{add}(w)$ is the `bitcount` of $w$ in binary format (Figure 3a) and $\mathcal{C}_{mv}(w, w')$ measures the proximity of two weight values $w$ and $w'$. In this study, we adopted $\mathcal{C}_{mv}(w, w') :=$ $|\mathrm{sgn}(w)|w|^p - \mathrm{sgn}(w')|w'|^p|$. We choose $p = 1/2$ to discount the change in $\mathcal{C}_{mv}$ for larger value (assuming accuracy is less affected by the change when it is large). Intuitively, $\mathcal{C}_{add}$ encourages $\hat{\boldsymbol{W}}_{j,k}$ to be sparse in the binary format while $\mathcal{C}_{mv}$ keeps it close to the current weight $\boldsymbol{W}_{j,k}$ and $\lambda^{(l)}$ controls its balance (larger $\lambda$ induce more sparsity). Owing to the discrete structure of the weight space (e.g., 8 bit), the solution of (8) can be analytically computed for given $\lambda$ (Figure 3b-3c).

**Efficient sparse inference.** Once the training has been completed, the high-precision `mult-add` network is converted into a mathematically equivalent `add-shift-add` network for efficient inference. It is expected to be comprised of sparse `add` which consumes less energy.

### 3.3 WHICH IS EFFICIENT? WEIGHT-PRUNING OR BIT-PRUNING

An important question in this study is whether pruning the `add` (Bit-Pruning) is more efficient than pruning the `mult` (Weight-Pruning). When considering the dot-product between two $K$-dimensional, $M$-bit vectors, the energy consumption of sparse `add-shift-add` ($E_{\texttt{add-shift-add}}$) over that of sparse `mult-add` ($E_{\texttt{mult-add}}$) is calculated from Table 2:

Table 2: Number of operations for dot-product between $K$-dimensional $M$-bit vectors.

|  | mult-add | | add-shift-add |
| --- | --- | --- | --- |
|  | dense | sparse | sparse/dense |
| mult | $K$ | $\sigma_{\texttt{mult}} K$ | 0 |
| add | $K-1$ | $\sigma_{\texttt{mult}} K$ | $M\sigma_{\texttt{add}} K + M - 1$ |
| shift | 0 | 0 | $M-1$ |

$$\frac{E_{\texttt{add-shift-add}}}{E_{\texttt{mult-add}}} = \frac{(M\sigma_{\texttt{add}} K)E_{\texttt{add}} + (M-1)E_{\texttt{shift}} + (M-1)E_{\texttt{add}}}{(\sigma_{\texttt{mult}} K)E_{\texttt{mult}} + (\sigma_{\texttt{mult}} (K-1))E_{\texttt{add}}} \approx \frac{M E_{\texttt{add}}}{E_{\texttt{mult}}} \frac{\sigma_{\texttt{add}}}{\sigma_{\texttt{mult}}} \quad (9)$$

where $\sigma_{\texttt{mult}}$ and $\sigma_{\texttt{add}}$ are the ratio of nonzero elements in weight ($\boldsymbol{W}_j$) and the ratio of nonzero bits in its binary format ($\boldsymbol{B}_j$), respectively. $E_{\texttt{mult}}$, $E_{\texttt{add}}$, and $E_{\texttt{shift}}$ are energy consumption of corresponding operations and we assume $K \gg M$ in the last approximation. In the case of ASIC, we

Table 3: Experimental setup

|  | CIFAR-10 | CIFAR-100 | ImageNet |
|---|---|---|---|
| Network | ResNet18 | ResNet18 | ConvNeXt-B |
| Batch size | 512 | | 256 |
| Training epochs | 200 | | 100 |
| Optimizer | AdamW (Loshchilov & Hutter, 2019) | | |
| Scheduler | OneCycle (Smith & Topin, 2019) | | Cosine decay (Loshchilov & Hutter, 2017) |
| Weight quantization $M$ | 8 | | |
| Activation quantization $N$ | 4/8/32 | | 8 |
| Weight initialization | Kaiming-uniform (He et al., 2015) | | Pretrained[4] |
| $\lambda^{(l)}$ for $C_{move}$ (Bit-Pruning only) | 1.0 for all layers (Figure 3c) | | |

roughly get $E_{\mathtt{mult}} \approx M E_{\mathtt{add}}$ from Table 1; therefore, the computational efficiency of Bit-Pruning over Weight-Pruning is almost determined by the ratio of obtained connection density $\sigma_{\mathtt{add}}/\sigma_{\mathtt{mult}}$.

We expect that the $\sigma_{\mathtt{add}}$ in a Bit-Pruned network is much smaller than the $\sigma_{\mathtt{mult}}$ in a Weight-Pruned network when both networks achieve comparable accuracy. This is because Bit-Pruning removes connections more finely than Weight-Pruning; in other words, given the same nonzero ratio, i.e., $\sigma_{\mathtt{add}} = \sigma_{\mathtt{mult}}$, Bit-Pruned weight *can* have a more nonzero element (at most $M$ times) than Weight-Pruned weight when it learns a weight that is sparse in binary format. In the extreme case where all the nonzero weights in a network are represented as PoT, the computational cost in `add-shift-add` representation is $M$ times smaller than that in `mult-add` representation. The evaluation on actual DNN training is presented in Section 4.

### 3.4 BIT-PRUNING AS A UNIFIED LOW-ENERGY MODEL

**(Non-uniform) quantization.** Bit-Pruning effectively applies different non-uniform mixed-precision quantization for *each weight element*. As a result of the Bit-Pruning, we will get a sparse $B$; this has a similar effect to the non-uniform quantization of weight. Existing quantization techniques learn (non-uniform) step size for a *group of weights*; contrary, Bit-Pruning realizes pseudo-non-uniform quantization by *selecting a bit pattern*.

**PoT dot-product.** Bit-Pruning can be interpreted as the soft version of the (additive) PoT quantization. The bit-sparsity regularization prefers sparse binary representation, which corresponds to the PoT; however, it could learn to choose individual non-PoT values for each weight element when necessary (i.e., when it achieves less total energy for the same accuracy). The learned weight by Bit-Pruning is *dominated by PoT, while some have multiple nonzero bits* (See Figure 6).

**Unstructured Weight-Pruning.** Bit-Pruning can be seen as a soft, fine-grained version of Weight-Pruning. Both bit-sparsity and weight-sparsity regularization loss are minimized when $W = 0$; this corresponds to zero `add` in `add-shift-add` and zero `mult` in `mult-add`. However, bit-sparsity regularization does not necessarily remove the *whole weight elements* but only *some unnecessary bits* in the weight element.

## 4 EXPERIMENT

### 4.1 EVALUATION OF ACCURACY/ENERGY TRADE-OFF

*Which is energy efficient; pruning `add` (Bit-Pruning) or pruning `mult` (Weight-Pruning)?* We evaluated the accuracy/energy trade-off of both methods on CIFAR-10 (Krizhevsky et al., a), CIFAR-100 (Krizhevsky et al., b), and ImageNet (Deng et al., 2009). For both bit-sparsity regularization $\mathcal{L}_{bit}$ (7) and weight-sparsity regularization $\mathcal{L}_{wgt}$ (3) which is used for all the conv layer, we approximate $l_0$-norm using Hoyer-Square loss (Yang et al., 2020). We use the same network architecture, quantization method (we use LSQ Esser et al. (2020)), and training strategy except for the loss function for both frameworks. The primary experimental setup[5] is summarized in Table 3. To compare the trade-off, we train networks across various sparsity coefficients $\eta$ of $\mathcal{L}_{wgt}$ and $\mathcal{L}_{bit}$ (fixed across the entire training sequence).

---

[4]Pytorch official model zoo http://pytorch.org/vision/main/models/resnext

[5]See appendix for more detailed experimental setup (Appendix A) and additional results (Appendix B).

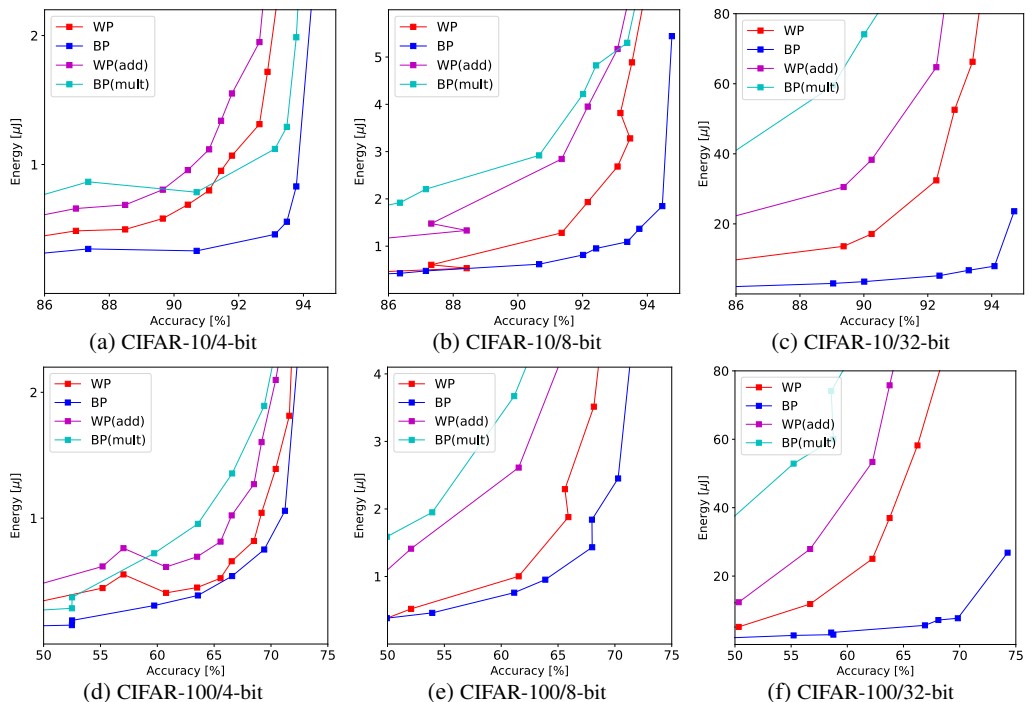

Figure 4: Accuracy/energy trade-off of ResNet18 trained on CIFAR-10 and CIFAR-100 for difference activation quantization level. Bit-Pruning (`add-shift-add`) (BP) and Weight-Pruning (`mult-add`) (WP) are compared. Bit-Pruning always uses an 8-bit width binary tensor $B$, and Weight-Pruning quantizes weight using the same quantization level for activation. The result of Weight-Pruning in `add-shift-add` representation (WP(add)) and Bit-Pruning in `mult-add` representation (BP(mult)) is provided for reference. The multiply-accumulate (MAC) of the original dense `mult-add` network is about $0.55 \times 10^6$, and the estimated energy consumption (from Table 1) are $26\mu$J, $107\mu$J, and $1721\mu$J for 4-bit, 8-bit, and 32-bit network, respectively.

**CIFAR-10/CIFAR-100.** We use ResNet18 (He et al., 2016a) as one of the most popular architectures and train them from scratch. Figure 4 summarizes the results on CIFAR-10 (Figure 4a, 4b, 4c) and CIFAR-100 (Figure 4d, 4e, 4f).

**ImageNet.** We conducted an experiment using pre-trained ConvNeXt (Liu et al., 2022) on ImageNet (Deng et al., 2009). The ConvNeXt is designed to be energy efficient by its network architecture; c.f., it heavily utilizes depthwise convolutions to reduce the number of `mult`. Results are summarized in Figure 5.

**Summary.** In all ranges of accuracy, `add-shift-add` network trained with Bit-Pruning consumes less energy than that of `mult-add` network trained with Weight-Pruning. In particular, huge energy saving is observed in high accuracy regime, demon-

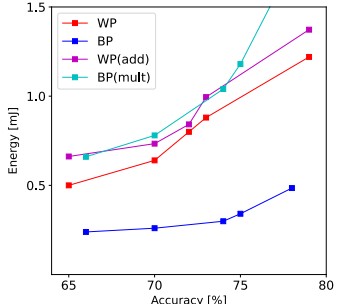

Figure 5: Result of ConvNeXt-B on ImageNet. 8bit activation. See Figure 4 for legend.

strating the superior data-adaptivity of Bit-Pruning over Weight-Pruning. We see a similar trend for the large-sized, already efficient networks (ConvNeXt) on a large-scale dataset (ImageNet). The number of `add` (in `add-shift-add` network) and `mult` (in `mult-add` network) can be compared by multiplying $M$ (number of bits for weight) to the `add` count in the plots (Because we have $E_{\texttt{mult}} \approx ME_{\texttt{add}}$ from Table 1, Section 3.3). The number of `add` (from Bit-Pruning) is larger than `mult` (from Weight-Pruning), however; their ratio is significantly less than $M$; therefore, we'll get the better trade-off.

Notice that the Bit-Pruned network does not show efficiency in `mult-add` representation (BP(mult)), and the Weight-Pruned network does not show efficiency in `add-shift-add` representation (WP(add)). This is because the Bit-Pruned network prefers sparse weights in binary format (Figure 6d), which is not necessarily zero, contrary, the Weight-pruned network prefer small weight, which is not necessarily sparse in binary format (Figure 6c).

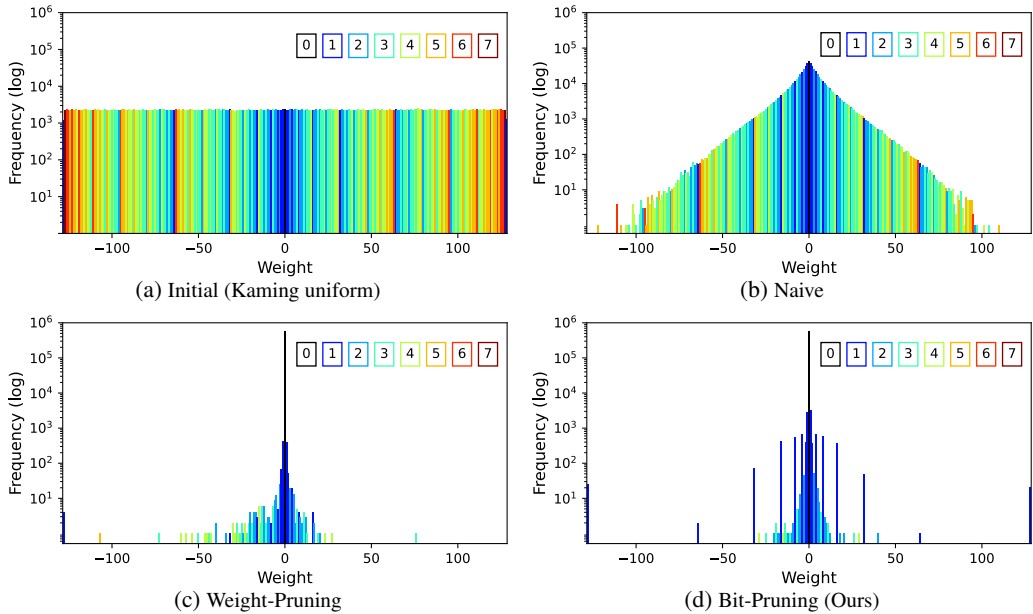

Figure 6: Weight histogram (CIFAR-10/ResNet18 13th convolution layer). *Naive* is a model trained using only task loss (test accuracy: 93.1%). We choose a model performing around 93.0% test accuracy for Weight-Pruning and Bit-Pruning. The color indicates the number of addition for `add-shift-add`.

## 4.2 ABLATION ON BASE BIT WIDTH

*Can we obtain a better accuracy-energy trade-off by pruning bits from a wider bit tensor?* We conducted the accuracy/energy trade-off comparison between Bit-Pruning with different base bit widths (4-bit/8-bit) (Figure 7). The result shows the network with a wide tensor (8-bit) shows a better trade-off than one using a narrow tensor (4bit). We consider it is because a network with a wide bit has more choices for selecting *good* PoT values, suggesting that a better bit-sparse network is obtained from the larger design space, just like the case of Weight-Pruning (Frankle & Carbin, 2018).

## 4.3 LEARNED WEIGHT DISTRIBUTION

Figure 6 visualizes histograms of learned weight. Starting from the initial uniform distribution, both Bit-Pruning and Weight-Pruning drive most of the weight to zero; this corresponds to removing `mult` in the `mult-add` network and `add` in the `add-shift-add` network, respectively. In Bit-Pruning, most of the remaining nonzero weights are concentrated on sparse values in a binary format, such as PoT. Whereas, Weight-Pruning did not show such particular preference.

## 5 RELATED WORKS

**ShiftAddNet** (You et al., 2020) is `mult`-less DNNs utilizing `shift` and `add` for efficient training and inference. It realizes a better accuracy/energy trade-off than the former study, AdderNet (Chen et al., 2020), utilizing only `add`. Ours and ShiftAddNet have similarities in that both utilize `shift` and `add` to realize `mult`-less DNNs. However, their computational model is fundamentally different from ours; ShiftAddNet computes the interaction between weight and input by first computing the dot-product with PoT weight followed by the $l_1$ norm with `add` weight. It does not have an equivalent `mult-add` dot-product. Its limited expressiveness and the non-differentiability of the PoT operation make training harder (requiring specifically designed gradient computation) and prohibit it from reaching the accuracy of the corresponding `mult-add` network. Ours can be learned as an ordinal high-precision differentiable `mult-add` network; therefore, it is possible to reach the performance of the original network. Besides, motivation and target hardware is different; ShiftAddNet aims to realize efficient training and inference on dense vector-type processors, while

our target is hardware supporting unstructured sparsity solely for inference[6]. Instead, using the recent extension, ShiftAddNAS You et al. (2022), ours and ShiftAddNet can be combined, e.g., the best combination of `shift-add` and `add-shift-add` are automatically searched.

**Lottery ticket hypothesis** states that *"dense, randomly-initialized, feed-forward networks contain sub-networks that have the equivalent accuracy as the original network (winning tickets): unstructured weight pruning naturally uncovers the winning tickets"* (Frankle & Carbin, 2018; Ramanujan et al., 2020). The comparison of different bit widths in Bit-Pruning (Section 4.2) might indicate the *lottery ticket hypothesis* in *bits*; a network having more connections in terms of `add-shift-add` representation (i.e., the large binary tensor $B$) has more chance of containing the *winning ticket*.

**Bit-level sparsity** have been considered in several areas. BSQ Yang et al. (2021) proposed a novel technique for learning the layer-wise bit-width for mixed-precision networks by inducing grouped bit-label sparsity. The novel notion of bit-slice sparsity (sparsity within a sliced subdivision of binary weight) is introduced by Zhang et al. (2019) to realize efficient inference on emerging ReRAM-based DNN accelerators. Because their motivation for considering the bit-level sparsity differs from ours, their proposed method for inducing it is also totally different. Therefore, it cannot efficiently sparsify the `add` in the `add-shift-add` network[7].

# 6 CONCLUSION

We propose Bit-Pruning, a novel framework for learning efficient `mult`-less sparse DNNs. The intensive evaluation shows pruning `add` (bit) of the `add-shift-add` network is more energy efficient than pruning `mult` (weight) of the `mult-add` network.

## 6.1 LIMITATIONS & FUTURE WORK

**Learning weight moving cost.** In Bit-Pruning, sparse `add` for efficient inference is achieved by guiding the weights closer to the proximal weight (sparse in binary format and close to the current weight) defined in (8). The distance to current weight needs to be defined as weight moving cost $\mathcal{C}_{mv}$. In this study, we predefined it (Section 3.2). We consider this to be sub-optimal, and it might be possible to determine the cost (e.g., by learning $p$) from data to achieve a better tradeoff, e.g., some weight can move a lot without affecting the task loss and vice versa.

**Evaluation on quantization susceptible module.** In this study, we evaluate Bit-Pruning on simple classification tasks using simple network modules such as `conv`. Some DNN modules are known to be more susceptible to quantization noise, such as the object detection head (Li et al., 2019a), transformer module

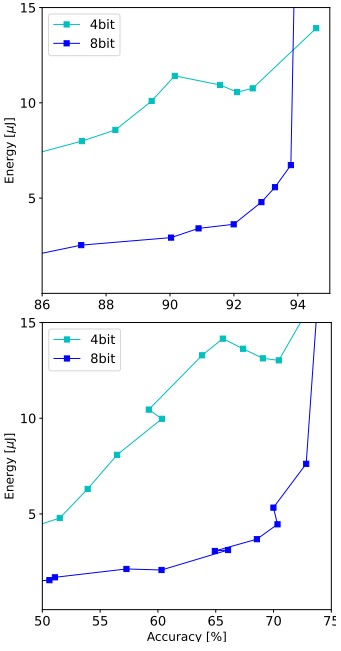

Figure 7: Ablation of bit width. CIFAR-10/100 (top/bottom).

(Bamberg et al., 2023), and recurrent module (He et al., 2016b), making it challenging to quantize them in a lower bit regime. Bit-Pruning might be an ideal choice for realizing energy-efficient inference on these modules because it could virtually utilize high-precision weight to achieve the accuracy of the original high-precision network.

**Benchmark on real hardware** We show the estimate of computational gain by Bit-Pruning using the basic statistic known in the literature (Table 1); we have not evaluated the actual energy consumption, wall-clock time, or area cost in the actual hardware. The proposed `add-shift-add` or existing `sparse mult-add` models require specialized hardware that supports unstructured/dynamic sparsity or optimized CPU implementation Kurtz et al. (2020) to exploit the sparsity to the full extent. The sparsity-aware accelerators often utilize in/near-memory computing architecture (Gholami et al.). We believe the Bit-Pruning is more suitable than Weight-Pruning for this architecture because `add` logic is cheaper in size and energy to place in or near memory than `mult` logic.

---

[6]For this reason, it is difficult to compare with ShiftAddNet in terms of computational efficiency.

[7]Refer to Appendix-E,H for more discussion and experiments for comparing the ability to induce sparsity.

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
