# OpenReview forum: "Bit-Pruning: A Sparse Multiplication-Less Dot-Product"
_ICLR.cc/2023/Conference — ICLR 2023 poster_

### Official Review · Reviewer_79uo · 2022-10-23

**Confidence:** 5
**Correctness:** 3
**Technical Novelty And Significance:** 3
**Empirical Novelty And Significance:** 3
**Recommendation:** 6

**Clarity, Quality, Novelty And Reproducibility:**

## Novelty
This paper is novel in the aspect that it motivates the use of random bit-level sparsity, and propose a regualrization method to induce such sparsity. However, this novelty is incremental since bit-level pruning has been explored in previous work (e.g. [1] [2]), as well as utilizing shift add operations (e.g. [3]). Also the paper does not make direct comparison with these previous explorations, which further doubts the significance of the proposed approach.

[1] https://arxiv.org/pdf/2102.10462.pdf

[2] https://arxiv.org/pdf/1909.08496.pdf

[3] https://proceedings.neurips.cc/paper/2020/file/1cf44d7975e6c86cffa70cae95b5fbb2-Paper.pdf

## Clarity
The paper clearly explains why bit sparsity and add-shift-add operation can bring additional energy savings. However, the paper does not give a clear explaination on why the proposed regularization in Eq. (6) is effective. For instance:
1. Why is $X^{(l)}$ involved in the weight regularization formulation?
2. Why is square root needed in the formulation of $C_{mv}$?
3. Why can the proposed regularization guarantees to induce bit sparsity? For example the weight value of 15 and 17 would have the same proximal $\hat{W} =16$, and have the same regularization value, but obviously 15 more 1-bits than 17. How is such situation avoided by the proposed regularization?

## Quality
The paper provides multiple experiments comparing the energy-accuracy tradeoff of BP vs. WP. However, the experiments are not well designed to ensure a fair comparison. For example:
1. Even under WP, nonzero weight also contains zero bits, which can also lead to energy saving if add-shift-add scheme is used. The paper ignores this in the computation and assumed no bit sparsity in the WP model
2. Quantizing the model to a lower precision naturally leads to less bits, thus less 1-bits. So its inadequate to only compare the proposed method with WP, but also to quantization to a lower precision.
3. Given the random access pattern of 1-bits in the model after bit prune, it would require specialized data pipeline to fulfill zero skipping in the add-shift-add operation, and would lead to additional overhead for data loading and storing. These potential overheads are not clearly discussed in the paper

**Strength And Weaknesses:**

## Strength
1. The paper provides interesting observation on the potential energy consumption improvement brought by add-shift-add operation
2. The paper advocates the exploration of utilizing bit-level sparsity, which may inspire future work

## Weakness
1. From the novelty perspective, the paper ignores previous attempts on inducing bit sparsity, and didn't perform comparison on closly related work. See the novelty section in the next review question for details
2. The paper does not clearly explain why the proposed regularization is effective. See clearity section of the next review question
3. The paper does not perform fair comparison in the experiments, and miss discussion on important baseline and limitation. See quality section in the next review question for details

**Summary Of The Paper:**

This paper considers the potential energy saving of implementing multiplication as a add-shift-add operation, and propose to utilize the bit-level sparsity of the weight to further reduce energy cost. The paper proposes a bit-sparsity regularization that promote weight to be sparse in binary format, and claim to achieve a better energy-accuracy tradeoff.

**Summary Of The Review:**

In summary, the paper makes progress on the exploration of bit sparsity, and how to utlize it to improve energy efficiency. However, the missing discussion on closely related work, the lack of clearity in the proposed method, and the unfair experiment setting make it hard to evaluate the significance and true effectiveness of the proposed method. Also, the potential overhead of the add-shift-add scheme may also hinders the real application of the proposed random bit sparsity. Therefore I would recommend rejection for now.

## Post rebuttal

I would like to thank the author for the extensive responses. Overall I like the idea of exploring potential energy savings of the random bit-level sparsity via the add-shift-add scheme. Also, the additional comparison provided by the author on using add-shift-add on weight pruned model indicates improving efficiency with add-shift-add is not naive for any model, and the explicit exploration of bit-level sparsity is required. This enhances the significance of the proposed regularization method.

As for the proposed regularization-based training method, the response from the author largely resolves my doubt on the effectiveness and the design choices. I think this paper should be worth acceptance given all the additional results and explorations provided by the author. To this end I'm increasing my score to weak accept.

---

> ### Author Response · Authors · 2022-11-19
> **Authors Response to Reviewer 79uo (1/5)**
>
> We thank the reviewer $\textbf{79uo}$ for clearly pointing out strengths and weaknesses.
> We especially appreciate the information about the related literature we did not know and the comments about the technical detail that can be clearer.
> By addressing the valuable comments (with new experiments), we believe the latest manuscript is clear and provide meaningful information to the community.
> In the following paragraph, we clarify the concerns raised by the reviewer.
>
>
> >1. From the novelty perspective, the paper ignores previous attempts on inducing bit sparsity, and didn't perform comparison on closely related work.
> This novelty is incremental since bit-level pruning has been explored in previous work (e.g. [1] [2]), as well as utilizing shift add operations (e.g. [3]).
>
> We appreciate the information about the related work,  which we failed to include in the original manuscript.
> Thanks to this comment, we find the positioning of this study can be more explicit.
> Our goal is to find the sparse $\operatorname{add}$ in the proposed $\operatorname{add-shift-add}$ formulation targeting for the architecture supporting the unstructured sparsity (e.g., optimized CPU implementation such as NeuralMagic, specifically designed DNN accelerator for sparse processing, FPGA, etc.).
> We induce the bit-level sparsity by minimizing the bit-sparsity loss, defined as the difference with the proximal weight $\hat{\boldsymbol{W}}$.
> Our contribution is not the introduction of the notion of bit-level sparsity but the framework for realizing $\operatorname{mult}$-free DNN based on the proposed $\operatorname{add-shift-add}$ formulation and the optimization method of the binary parameter in the continuous weight space using the proximal weight.
> We've clarified the above point in Sec.1 and Sec.3, and we added the summary of the following discussion about the literature provided by $\textbf{79uo}$ in Sec.5 (we've added a new paragraph for this discussion).
> Furthermore, we've conducted a new experiment comparing with bit-slice sparsity of [2] in terms of their ability to induce bit-level sparsity in our $\operatorname{add-shift-add}$ formulation.
> And we've added the new results and following discussion about their mechanism for inducing the sparsity in Appendix.
>
> > 1.1 BSQ
>
> BSQ [1] proposed a novel method for realizing gradient-based optimization of the layer-wise precision,  targeting the hardware supporting the mixed precision.
> They formulate the optimization of layer-wise precision as the bit-wise sparsification.
> They represent the weight using the sum of the PoT basis (same as our binary representation of Eq.4).
> Then they directly optimize for each bit using a straight-through estimator (STE).
> Ours and BSQ [1] differ in their motivation and technique for inducing bit-level sparsity.
> The goal of BSQ is to learn the layer-wise precision targeting the hardware supporting the dense mixed-precision operation (e.g., GPUs); on the other hand, our goal is to realize an efficient $\operatorname{mult}$-free network based on the proposed $\operatorname{add-shift-add}$ with sparse $\operatorname{add}$ targeting the hardware supporting the unstructured sparsity (e.g., data-flow processors, CPUs, FPGAs, etc.).
> Due to the difference in their goal, they differ in their mechanism for inducing the sparsity; BSQ [1] applies the sparsification for *groups of weights* (e.g., each layer), while we apply the sparsification for each *individual weight element*.
> Furthermore, in BSQ,  sparsity is induced by *optimizing each bit independently* using STE, which may suffer from severe quantization error.
> In contrast, ours *optimize in the quasi-continuous weight space* using the proximal weight based on the novel notion of the equivalence of the  $\operatorname{add-shift-add}$ and $\operatorname{mult-add}$.
> We've clarified this point in the revised manuscript.
>
> > 1.2 ShifAddNet
>
> We discuss the relation to ShifAddNet [3] in Sec.5.
> We've updated the section by adding the discussion with the ShiftAddNAS (kindly informed by $\textbf{bvLy}$).
> We did not compare the accuracy/energy tradeoff with them due to the different motivations and architecture, which makes the direct comparison difficult ($\textbf{bvLy}$ also explains this point).
> Please also refer to the related discussion from other reviews ($\textbf{bvLy}$-(4))

---

> ### Author Response · Authors · 2022-11-19
> **Authors Response to Reviewer 79uo (2/5)**
>
> > 1.3 Bit-slice sparsity
>
> The study of [2] (which we did not know) proposed the concept of bit-slice sparsity.
> Their target hardware is novel ReRAM-based accelerators, where each crossbar can not hold the whole 8-bit weight; therefore, slices of the binarized weight element (bit-sliced weight) are distributed into the different crossbars.
> In this hardware restriction, their goal is to sparsify each slice.
> To this end, they proposed the bit-slice $l_1$ regularization, defined as a $l_1$ norm of each sliced weight value (in their case, each slice has 2bit).
> By reducing the  $l_1$ norm for each slice, the weight as a whole is also expected to have fewer nonzero elements in binary representation.
>
> Ours and  [2] are different in motivation; therefore, their technique for inducing bit-sparsity is also different.
> Their goal is to induce the sparsity *within the slice*; on the hand, our goal is to induce the sparsity in the binarized weight as a whole.
> They induce the sparsity by minimizing the $l_1$ loss within each slice and do not consider the continuity of weight as a whole; on the hand, bit sparsity regularization takes the continuity of the whole weight into account when inducing the sparsity to keep the change of weight value minimum to prevent the accuracy deterioration by the significant change.
>
> Although the motivation is different, Bit-Pruning and bit-slice sparsification of [2] share some similarities in the scene in that both methods aim at realizing the sparse representation of weight in binary format.
> Therefore, we conducted additional experiments by comparing them in terms of their ability to induce sparsity.
> We added the visualization of the loss landscape of their bit-slice regularization $\mathcal{L}_{bitslice}$ in Fig.18.
> Because the $l_1$ loss is computed inside the slice, it can not take the state of other bits into account; therefore, it can not take the continuity of the weight into account.
> When the number of slices is 1 (Fig.18-(d)), the  bit-slice $l_1$ loss reduces to the $l1$ regularization for the weight value; therefore, we can not expect the sparsity because it simply reduces the norm of the weight.
> When the number of slices equals the bit width $M$ (Fig.18-(a)), it guides each bit to zero, ignoring the weight value's continuity.
> When the number of slices is 2 (Fig.18-(c), which is their experimental setup), it only considers the continuity in the slice and ignores the continuity as weight as a whole.
>
> For example, consider the case when $w=63$ (6 nonzero bit), then our proximal weight is $w=64$  (1 nonzero bit).
> Weight ($w=63$) is guided toward the proximal weight, which differs only 1 in the weight value.
> Still, nonzero bits are reduced significantly (from 6 to 1).
> In case of the  bit-slice $l_1$ loss it guides the weight towards $63_{10}=(00|11|11|11)_{2}\mapsto 42=(00|10|10|10)$.
> The corresponding weight value changes drastically; therefore, accuracy may drop significantly, but the gain for the bit sparsity is small (from 6 to 3).
> Our bit-sparsity loss guides the weight that maximizes the gain while preventing the weight from changing a lot by simultaneously considering the proximity of the weight value and the number of bits.
>
> We conducted new experiments to compare our bit-sparsity regularization and their bit-slice $l_1$ regulation [2].
> We modified their loss function `calc_l1_and_zero_ratio` from their provided code (https://github.com/zjysteven/bitslice_sparsity) to be used in our Bit-Pruning framework.
> For a fair comparison, we also use $l_1$ loss (instead of Hoyer-Square loss) for our bit-sparsity regularization.
> We added the result in Fig.19 and compared it with our results.
> Our bit-sparsity regularization ($l_1$ version) shows a better tradeoff than the bit-slice $l_1$ regulation.

---

> ### Author Response · Authors · 2022-11-19
> **Authors Response to Reviewer 79uo (3/5)**
>
>
> >2. Why is $X$ involved in the weight regularization formulation?
>
> We include $X$ in our bit-sparsity regularization (Eq. 7) and weight-sparsity regularization (Eq. 3) because most DNN accelerators supporting the unstructured weight sparsity usually equip the functionality to utilize the sparsity in activation.
> And we want to compare the method in this hardware setup.
> Optimized CPU implementation for unstructured weight sparsity, such as by NeuralMagic, also supports activation sparsity (for Intel-based CPUs).
> In the case of Weight-Pruning, the computation is skipped either when $\boldsymbol{X}$ or $\boldsymbol{W}$ is sparse  (Sec. 2.2).
> In the case of Bit-Pruning, we can similarly skip the computation of  $\operatorname{add}$ when the corresponding $\boldsymbol{X}$ is zero.
> We've clarified this point in Sec. 2.2 and Sec. 3.2.
> And we also clarified the discussion when hardware only supports the unstructured weight sparsity as follows.
>
> When we consider the hardware which only supports unstructured weight sparsity, the regularization for Weight-Pruning and Bit-Pruning is defined by replacing $\boldsymbol{X}^{(l)} $ with  $\operatorname{numel}(\boldsymbol{X}^{(l)})$.
>
> Our code which will be open-sourced has options to enable/disable the activation sparsity.
>
> >3. Why is square root needed in the formulation of $\mathcal{C}_{mv}$
>
> We use $p=1/2$ because we consider the changes of weight value when its norm is small to have more effect on the accuracy than when its norm is large.
> For example, changes of 2 when the weight was 1 (200\% change) may have a more significant impact on accuracy than the weight was 200 (1\% change).
> When $p=1/2$, the weight moving cost $C_{mv}$ changes a lot when its norm is small and does not change much when its norm is large.
> We've clarified this point in Sec. 3.1.
>
> Furthermore, thanks to the $\textbf{79uo}$'s comments, we noticed the ablation in this parameter is very informative to show how this choice affects the accuracy/energy tradeoff.
> Therefore, we conducted a new ablation experiment using the moving cost $C_{mv}$ with $p=1.0$.
> This choice assumes the change in weight will equally affect the accuracy regardless of the norm of the current weight.
> We've added the visualization of the loss landscape  $\mathcal{L}_{bit}$ with $p=1.0$ in Fig.17.
> It is significantly different from the case of  $p=1/2$ shown in Fig.11.
>
> Fig.16  shows the new results when $p=1.0$.
> We observe a degradation in the accuracy/energy tradeoff.
> We suspect this is because the  $\mathcal{L}_{bit}$ with   $p=1.0$ does not reflect the tradeoff of changing the value of weight to accuracy.
>
> This result suggests that we could expect better accuracy/energy tradeoff by using *smarter* $C_{mv}$, e.g., by learning from data (as we discussed in future work in Sec.6).
>
>
> >4. Even under WP, nonzero weight also contains zero bits, which can also lead to energy saving if $\operatorname{add-shift-add}$ scheme is used.
>
> Exactly.
> The learned weight using the Weight-Pruning scheme could also enjoy bit-wise sparsity when it is converted into the $\operatorname{add-shift-add}$ format.
> Thanks to the $\textbf{79uo}$'s comments, we find providing these results are very informative to better compare with the Weight-Pruning.
> Therefore, we've added the result for all the plots in Fig. 4 and 5 (WP(add)).
>
> As expected, the accuracy/energy tradeoff of the weight-pruned model in $\operatorname{add-shift-add}$ mode is worse than that of the bit-pruned model in $\operatorname{add-shift-add}$ mode.
> The network trained with the weight sparsity regulation $\mathcal{L}_{wgt}$ (Eq.3) is optimized to induce weight-level sparsity and does not consider the bit-level sparsity; therefore the learned weight has many non-PoT values, as inspected from Fig.6 (c).
>
> The opposite is also true; when the network trained with the bit-sparsity regularization (Eq.6) is evaluated in the $\operatorname{mult-add}$ mode (BP (mult)), its tradeoff was worse than the network trained using the weight-sparsity regularization (Eq.3).
> This result also conforms to the learned weight histogram (Fig.6 (c)-(d)), where the number of zero weights from bit-sparsity regularization is less than that of weight-sparsity regularization.
> (The results of  BP(mult) had been presented in the original plot, we added the plot of WP(add))

---

> ### Author Response · Authors · 2022-11-19
> **Authors Response to Reviewer 79uo (4/5)**
>
>
> >5. Why can the proposed regularization guarantees to induce bit sparsity? For example the weight value of 15 and 17 would have the same proximal $\hat{\boldsymbol{W}}=16$, and have the same regularization value, but obviously 15 more 1-bits than 17. How is such situation avoided by the proposed regularization?
>
> Our Bit-Pruning loss is defined as the difference (in Hoyer-Square sense) between the proximal weight $\hat{\boldsymbol{W}}$ and current weight ${\boldsymbol{W}}$.
> The proximal weight takes both bit-cost and proximity to the current weight into account (Eq. 8).
> It is a compromised point (found by solving the optimization of Eq.8) that reduces the bit-cost while keeping the change from the current weight small as possible.
> The bit-cost $C_{bit}$ of the proximal weight is always less than or equal to the current bit-cost regardless of the choice of the moving cost $\mathcal{C}_{mv}$; therefore, sparsity is always induced.
>
> However, as pointed out by $\textbf{79uo}$, the bit sparsity regularization does NOT take the possible reduction of the bit-cost $C_{bit}$ into account once the proximal weight is determined.
> Thanks to $\textbf{79uo}$, we found the discussion on this point can be more precise, and we've clarified in  Sec.3, conducted new experiments regarding these points, and added the following discussion in Appendix G.
>
> Let's consider a similar case raised by $\textbf{79uo}$, $w=63$, and  $w=65$.
> They would have the same proximal weight,  $\hat{w}=64$.
> Obviously, $w=63$  has more nonzero bits than $w=65$.
> Using the bit-sparsity regularization (Eq.7), both $w=63$ and  $w=65$ are guided toward $w=64$, which is sparse in binary representation (only one nonzero bit).
> But we'll expect more gain from $w=63$ ($6\mapsto 1$) than $w=65$ ($2\mapsto 1$).
> The bit-sparsity regularization (Eq.7) does not take this into consideration.
> And as pointed out by $\textbf{79uo}$, they are treated equally, ignoring their different gain in bit-cost.
>
> We may get a better tradeoff by directly reflecting the gain into the loss function.
> To this end, we consider the weighted version of the bit sparsity regularization  defined as follows:
>
> $\mathcal{L}_{bit}(\mathcal{W}) =$
>
> &emsp; $\sum_{l=1}^L|\boldsymbol{X}^{(l)}\odot (\boldsymbol{W}^{(l)}-\hat{\boldsymbol{W}}^{(l)})|_{0}$
>
> $\mathcal{L}_{bit\ weighted}(\mathcal{W}) = $
>
> &emsp; $ \sum_{l=1}^L|\boldsymbol{X}^{(l)}\odot (\boldsymbol{W}^{(l)}-\hat{\boldsymbol{W}}^{(l)})(C_{add}({\boldsymbol{W}^{(l)})}-{C_{add}(\hat{\boldsymbol{W}}^{(l)})})|_{0},
> $
>
> We've added the new results using the weighted version of bit-sparsity regularization in Fig.20.
> Opposed to our expectations, we could not observe a noticeable improvement in the accuracy/energy tradeoff.
> We have not found the reason yet; we may observe some improvement using a different optimizer, model designs, or datasets.
> However, because our focus in this study is to compare Bit-Pruning and Weight-Pruning in a simple and fair setup; therefore, we left further explanation as future work.
>
>
> >6. Quantizing the model to a lower precision naturally leads to less bits, thus less 1-bits.
>
> We agree with $\textbf{79uo}$ on this point.
> Lower precision networks require less energy.
> We did not directly compare with the extremely low-bit model for the reason we discuss in the following.
> We've clarified the manuscript so that the goal and scope of the study are more precise.
>
> (1) Although quantization could reduce energy consumption, they *define* the energy-efficient model structure *before training*.
> Our motivation for developing the Bit-Pruning is to *learn*  efficient model structure *during training* (instead of pre-define) by removing unimportant *bit* just like Weight-Pruning removes unnecessary weights from high-capacity high-precision models.
> In this regard, i.e., pre-defined vs. learned during training, ours and quantization are orthogonal and difficult to compare directly.
>
> (2) Our goal is to *learn* efficient model which has FP32 comparable accuracy; however, quantized network in the low-bit regime impose training with a precision that is difficult to use gradient-based optimization (Sec.1), e.g., approximate gradient with the straight-through estimator (STE), making it challenging to reach that accuracy (it is difficult for a moderate 4bit model, refer to Fig.5 (e) and (f)).
> It may be possible to reach the FP32 accuracy by increasing the model capacity, such as by increasing the channel width; we consider this is outside the scope of this study.
>
> (3) For a fair comparison with the low-precision model, we need to compare with the low-precision model having unstructured sparsity.
> However, the unstructured low-precision model is a less explored area; furthermore,  several techniques (e.g., gradient computation for the quantization layer) are required to successfully train the low-bit network making the fair comparison even harder.

---

> ### Author Response · Authors · 2022-11-19
> **Authors Response to Reviewer 79uo (5/5)**
>
> >7. Given the random access pattern of 1-bits in the model after bit prune, it would require specialized data pipeline to fulfill zero skipping in the $\operatorname{add-shift-add}$ operation, and would lead to additional overhead for data loading and storing.
> These potential overheads are not clearly discussed in the paper
>
> We agree with $\textbf{79uo}$ that there will be overheads regarding the random memory access.
> Therefore, DNN accelerators supporting unstructured sparsity or optimized CPU implementation are necessary to realize the actual gain.
> Regarding this point, we've cleared the scope and goal of this study in Sec.1 and Sec.3.
>
> Recent advances in the in-memory computing architecture realized efficient matrix multiplication with weight having unstructured sparsity.
> The in-memory architecture mitigates the overhead regarding memory access by avoiding the energy-hungry random access to the eternal DRAM.
> Several accelerators supporting unstructured sparsity are now widely available (for R\&D purposes).
> General purposes CPU can also utilize the unstructured sparsity.
> For example, NeuralMagic provides a solution (called Tensor Columns) for the efficient execution of the network having unstructured weight sparsity (and activation sparsity) by intelligently utilizing the L1/L2 cache mechanism for Intel-based CPUs.
> Considering the reported gain by the Tensor Columns, we expect an actual reduction in energy will be at least a few dozen percent from the theoretical gain shown in Fig.5-6  (for both  Weight-Pruning and Bit-Pruning).
> We can expect more in the case of specifically designed accelerators or FPGAs.
>
> We are currently working on evaluating the Bit-Pruning on the CPUs by utilizing a similar technique as the NeuralMagic.
> We are also working on the evaluation using the data-flow type processors based on the in-memory computing architecture.
> The evaluation of  wall clock time and the actual energy consumption using the CPU and the dedicated processors is left for future work (Sec.6.1).
>
> Please also refer to the related discussion from other reviews ($\textbf{bvLy}$-(3), $\textbf{MKH6}$-(1)).

---

> ### Comment · Reviewer_79uo · 2022-11-19
> **Thanks for the response**
>
> I would like to thank the author for providing the extensive responses. I think the significance and the effectiveness of the proposed method is much more clear after the additional explorations by the author. I'm therefore increasing my score.

---

### Official Review · Reviewer_MKH6 · 2022-10-24

**Confidence:** 4
**Correctness:** 4
**Technical Novelty And Significance:** 2
**Empirical Novelty And Significance:** 2
**Recommendation:** 6

**Clarity, Quality, Novelty And Reproducibility:**

The paper is well written and technically sound. The authors do a good job explaining how the dot product can be implemented as an add-shift-add transformation, and the impact this can have considering sparsification at the bit level.

One aspect of the training that is not clear to me, is whether you intend to train the whole network from scratch using the sparsification, or simply refine the resulting network?

**Strength And Weaknesses:**

The major weakness is that the experiments were not carried on real hardware, which could demonstrate the benefits of the approach.

The introduction of the loss function that helps with the discretization is well defined, however, there are no more details on the impact it has on the optimization. The proximal weight function defines a discretization that is not smooth, and when we take into account the optimization step size, we could see oscillatory behaviors during training.

The empirical evaluation does a good job comparing the simulated energy consumption, however, more details on the intended training use-case would be beneficial (see next section).

**Summary Of The Paper:**

In this paper, the authors present an energy saving approach that replaces dot product operations with an equivalent add-shift-add operation. In the context of neural networks with integer weights, the authors demonstrate how such a technique can save significant amounts of energy while controlling for the accuracy degradation.

The paper introduces a theoretical foundation based on the binary decomposition of the dot product operation. They then replace the costly multiplication operations by unrolling additions up to the length of the binary encoding (e.g. 8 bits), and then use shifts and additions to reconstruct the dot product exactly.

The savings come from pruning at the binary level, where the authors introduce sparsity in a way that reduces the number of computations to be carried on.

The authors then present an empirical evaluation on a simulator, as they did not have access to real hardware that could realize this kind of operations.

**Summary Of The Review:**

The authors propose an interesting paper that can be viewed as a further discretization not in terms of bit counts as it is common in the DL literature, but they go one step further and take energy consumption into account by means of replacing a core operation such as the dot product, with a simplified version implemented by additions and shift operations.

The paper raises some further questions: Are you training the whole network using your approach? If so, as you are training with the weights in full precision and simply adding a penalization in the form of a proximal weight, it is not clear to me what the benefits are of penalizing during the whole training history. Or do you train at high precision and then simply refine with your penalty?

Furthermore, a regularization term in the loss, does not imply that the regularization term will have a cost of zero, except potentially when this is a significant part of the loss function. Do you employ a dynamic regime to rescale your regularization? i.e., do you increase your $\eta$ parameter during training? or is it kept constant?.

---

> ### Author Response · Authors · 2022-11-19
> **Authors Response to Reviewer MKH6 (1/2)**
>
> We thank the reviewer $\textbf{MKH6}$ for clearly pointing out strengths and weaknesses; especially, we appreciate the detailed comments regarding the clarity of the construction of the loss function and training procedure.
> In the following paragraph, we clarify the concerns raised by the reviewer.
>
> >1. The experiments were not carried on real hardware, which could demonstrate the benefits of the approach.
>
> We have not evaluated the performance in actual hardware.
> Our goal in this study is to answer the following question  "Which pruning offers a better accuracy/energy tradeoff? $\operatorname{mult}$ in $\operatorname{mult-add}$ network (Weight-Pruning) or $\operatorname{add}$ in $\operatorname{add-shift-add}$ network (Bit-Pruning)" (Sec.1).
> For this purpose, we've established the generic framework for comparing their energy consumption using the  general specs known in the literature (Tab.1).
> Although we agree with $\textbf{MKH6}$ on the importance of the evaluation in real hardware, we believe the proposed experimental results provide sufficient information about the benefit of fine-grained bit-level pruning over coarse weight-level pruning.
> We've clarified the above discussion on the scope and the goal of this study in Sec.1.
>
> We are now working on evaluating Weight-Pruning and Bit-Pruning in real hardware using CPUs and dedicated DNN accelerators.
> We plan to publish the results as a follow-up study (please also refer to the related discussion from the other reviewers $\textbf{bvLy}$-(2),(3) and $\textbf{79uo}$-(4)).
>
> >2. The introduction of the loss function that helps with the discretization is well defined; however, there are no more details on the impact it has on the optimization.
> The proximal weight function defines a discretization that is not smooth, and when we take into account the optimization step size, we could see oscillatory behaviors during training.
>
> As $\textbf{MKH6}$ points out, the target weight (proximal weight $\hat{\boldsymbol{W}}$) changes depending on the current weight value.
> When $\lambda=1.0, p=0.5$, there are 13 local minimums, and when the weight takes a value on the boundaries, the proximal weight may oscillate, and the optimization might also oscillate.
>
> We showed the training dynamic in Fig.9 in Appendix B.
> Empirically we did now see such oscillatory behavior.
> We suspect this is because the weight does not stay on the boundary region as it is guided towards the local minimum of $\mathcal{L}_{bit}$, which is far from the boundaries (Fig.11-(c)).
>
> To further investigate the stability, we have conducted new experiments using different proximal weights having more local minimum.
> We've added the new experimental result using $\lambda=0.1$ in Fig.14 (see Fig.11-(b) for loss landscape).
> We also did not observe the oscillatory behavior; however, the tradeoff was worse.
> The degenerated tradeoff may be attributed to the too-conservative proximal weight, or it may come from the difficulty in training using the loss function that has too many local minima  (Fig.11-(b)).
>
> >3. Are you training the whole network using your approach?
> If so, as you are training with the weights in full precision and simply adding a penalization in the form of a proximal weight, it is not clear to me what the benefits are of penalizing during the whole training history.
> Or do you train at high precision and then simply refine with your penalty?
>
> We train the whole network using our approach (Bit-Pruning).
> For comparison, we also train the whole network using Weight-Pruning.
> The network is quantized with sufficiently higher bit-width (e.g., 32-bit or 8bit for weighs and activation), which we expect to yield the FP32 comparable accuracy.
> (In this regard, results on the low-accuracy regime and results using 4bit activation are not our main use case, we present the results purely for comparison.)
> We assume $\boldsymbol{X}$ and $\boldsymbol{W}$ for all the equations in the paper are quantized.
> We've clarified the manuscript on this point in Sec.1 and Sec.2.
> (Thanks to the comment, we noticed that representing the weight and activation using $\mathbb{R}$ are not appropriate, and we represent all the quantized parameters using $\mathbb{Z}$)

---

> ### Author Response · Authors · 2022-11-19
> **Authors Response to Reviewer MKH6 (2/2)**
>
> >4. The empirical evaluation does a good job comparing the simulated energy consumption, however, more details on the intended training use case would be beneficial.
> ...
> Whether you intend to train the whole network from scratch using the sparsification, or simply refine the resulting network?
>
> We consider that Bit-Pruning (and Weight-Pruning) could be used for both scenarios, training the network from scratch or fine-tuning the pre-trained network, depending on the application scenario.
> Thanks to the comment, we notice the results comparing both scenarios is essential.
> We've conducted new experiments using a pre-trained network on CIFAR10  to compare the two scenarios and added the new results along with the following discussion in Fig. 15 and Appendix E.3.
>
> In the main paper, we trained the ResNet18 network from scratch for CIFAR10 and CIFAR100 and fine-tuned the ConvNeXt network from a pre-trained network for ImageNet.
> We trained the network from scratch because we wanted to compare Weight-Pruning and Bit-Pruning in a fair and simple setting as possible.
> We were concerned the results may be affected by the pre-trained network.
> (We use a pre-trained network for ConvNeXt because we could not run large-scale training using a larger dataset such as ImageNet-22K.)
>
> Nonetheless, we agree with $\textbf{MKH6}$ that it is important to compare the performance of the pre-trained and training-from-scratch scenarios.
> To this end, we conducted experiments comparing the Weight-Pruning and Bit-Pruning in both scenarios on CIFAR10.
> We used the network trained with $\eta=0$ as a pre-trained network for both Weight-Pruning and Bit-Pruning.
> The new experimental results are added in Fig.15.
> The accuracy/energy tradeoff is almost the same on the high-accuracy regime and we training converged earlier than the training-from-scratch case.
> But we observe the instability in the low energy regime (when $\eta$ is large).
> We could not find the reason for the instability, but it might be attributed to the learning-rate scheduler designed for the training-from-scratch scenario (e.g., warm-up, etc.)
>
> In summary, we consider Bit-Pruning (and Weight-Pruning) could be used for both scenarios.
> But given the similar tradeoff (at least on the high-accuracy regime), it may be beneficial to use a pre-trained network when available.
>
>
> >5. A regularization term in the loss, does not imply that the regularization term will have a cost of zero, except potentially when this is a significant part of the loss function.
> ... ,i.e., do you increase your $\eta$ parameter during training? or is it kept constant?
>
> We use constant $\eta$ across the entire training sequence.
> We adopted this setting because we want to compare the Weight-Pruning and Bit-Pruning in a simple and fair configuration as possible.
> We've clarified the choice in the experimental setup (Sec. 4.1).
>
> We agree with $\textbf{MKH6}$ that some clever scheduling of the $\eta$ will improve the accuracy/energy tradeoff; however, we prioritize the comparison on simple and fair configuration, and we left the exploration of the dynamic scheduling of $\eta$ as future work.
>
> For additional information, at the beginning of this research, we tried the opposite strategy, e.g., annealing  $\eta$ during training (attenuating $\eta$ in sync with the learning rate for the optimizer).
> But we did not observe a noticeable gain.

---

### Official Review · Reviewer_Rw8U · 2022-10-25

**Confidence:** 3
**Correctness:** 3
**Technical Novelty And Significance:** 2
**Empirical Novelty And Significance:** 3
**Recommendation:** 8

**Clarity, Quality, Novelty And Reproducibility:**

The paper is well written, with sufficient background for those that aren't familiar with lower-level compute and HW considerations. I'm unsure about the novelty here (unstructured sparsity is, as the authors state, not so easy to accelerate unless custom HW is available -- I believe limited work has been done in this front as a result but I'm not super familiar with the Literature these days) but the method proposed is simple enough, well motivated, and seems to deliver very good results. The Authors state the code will be open-sourced, that's always a plus.

The above being said, the worthiness of Bit-purning is subject to the fact that an accelerator for unstructured sparsity can actually be implemented and yield the estimated accuracy-energy trade-off.

**Strength And Weaknesses:**

### Strengths
*    The proposed method is simple and seems to deliver very good results, specially at higher accuracy ratios, where weight pruning understandably struggles to zero-out weights without impacting on accuracy.
*    ImageNet results shows around 50% energy reduction, that's good.


### Weaknesses
*    Unstructured sparsity benefits from models being over-parameterised for the task (i.e. the task at hand is easy so a fair amount of weights can be discarded without impacting on accuracy). It would be interesting to see the performance of Bit-pruning when the simpler tasks (specially CIFAR-10) is done with a light-weight model, for example a small mobilenet.
*    What approach was used to perform weight-pruning? using traditional magnitude-based pruning? or something else?
*    Some key implementation details aren't clear from the text: are all convolutional layers using Bit-pruning? both 3x3 and 1x1 convolutions? Also the input layer?
* Why all layers use the same $\lambda$? Probably earlier layers would had benefit for less sparse weights retaining in that way some of the degradation that would otherwise propagate to the rest of the network. Another motivation for this is that early layers have comparatively much fewer OPs than deeper layers so the potential speedup might not be worth it. Could the Authors comment upon this?

**Summary Of The Paper:**

Bit-pruning proposes pruning at the bit level in the context of unstructured pruning methods. At its core, formulation presented by the Authors aims to replace the scalar-scalar multiplication with its equivalent scalar-Nbit multiplication followed by N-shifts and finally a sum over N shifted values. When extending this to a dot-product (the basic OP in matrix-matrix and matrix-vector multiplications), it can be seen as a add-shift-add OP. In this formulation to compute is dominated by the number of bit-level additions before the shift and therefore this is the metric to minimise. By means of an additional loss term that penalises the number of bits in the weights representation as well as the distortion this introduces, the proposed bit-pruning mechanism achieves large energy reductions for a fixed accuracy value. This method is evaluate on image classification with ResNets.

**Summary Of The Review:**

This is a good paper that shows how pruning at the bit level can yield much better energy-accuracy ratios than weight-pruning that prunes weights directly. Even when at lower bit-widths, bit-pruning (which uses 8-bit weights) outperforms the weight-pruning counterpart using 4-bit weights. I would had loved to see this method applied to smaller models (even if CNNs) or to Transformers (even if only a single ViT).

---

> ### Author Response · Authors · 2022-11-19
> **Authors Response to Reviewer Rw8U (1/2)**
>
> We thank the reviewer $\textbf{Rw8U}$ for understanding the study's contribution and pointing out the unclear explanation to strengthen the paper.
> In the following paragraph, we clarify the concerns raised by the reviewer.
>
> >1. Unstructured sparsity benefits from models being over-parameterised for the task ...
> It would be interesting to see the performance of Bit-pruning when the simpler tasks (specially CIFAR-10) is done with a light-weight model, for example a small MobileNet.
> I would had loved to see this method applied to smaller models.
>
> As $\textbf{Rw8U}$ explains, Wight-Pruning and Bit-Pruning assume some redundancy in weights to be pruned.
> We agree that the results when the model is small are interesting.
>
> To investigate the behavior of both methods in a situation where the number of possible unnecessary weights is scarce, we've conducted experiments using the narrow version of ResNet18 (narrow ResNet18).
> The narrow  ResNet18 has $8\times$ less input and output channels w.r.t the original ResNet18, except the input and final layer.
> In this situation, we can not expect significant pruning (either weights or bits).
> We added the new results in Fig.13 and the following discussion in Appendix E.1.
> As expected, even when there is no sparsity regularization, e.g., $\eta=0$, we already observe significant degeneration in accuracy compared with the original full-sized ResNet18, both on Weight-Pruning and Bit-Pruning.
> As expected, by increasing the regularization weight $\eta$, we observed the reduction in energy at the cost of accuracy degeneration on both algorithms.
> We still observed a similar trend as in the case of the full-sized network; the Bit-Pruned network still shows better accuracy/energy tradeoff.
>
> Regarding light-weight structure, our experiments on ImageNet use the ConvNeXt.
> The ConvNeXt is designed to be energy efficient by its network architecture; c.f., it heavily utilizes depthwise convolutions to reduce the number of $\operatorname{mult}$ (in this regard of using depth-wise convolution,  ConvNeXt is similar to MobileNet).
>
>
> >2. What approach was used to perform weight-pruning?
> ...
> Using traditional magnitude-based pruning?
>
> Yes.
> For Weight-Pruning, we use a magnitude-based algorithm using *Hoyer-Square* [Yang et al.] for the magnitude computation combined with the LSQ-based quantization.
> We've clarified this point in Sec.2.2 and Sec.4.1 and added the detail in Appendix A.4.
> In Weight-Pruning, weights elements that are zero after the quantization is pruned.
> The algorithm for the Weight-Pruning is similar to the one proposed in the *neural-wiring* by [Wortsman et al.], *edge-popup* by [Ramanujan et al.], and the ones adopted by *Hoyer-Square* [Yang et al.].
> But instead of the threshold for pruning, we use quantization by LSQ and weights elements that are zero after the quantization is pruned.
> Therefore, pruned weight can be active again, which is key to finding a good sparse network.
>
> We adopt this Weight-Pruning setting for a fair comparison with the proposed Bit-Pruning.
> In Bit-Pruning, quantized weight is converted into binary format, and zeros elements in the binary format are pruned.
> In other words, both methods prune a value smaller than the threshold (e.g., half the quantization scale) where the difference between the target weight ($\mathbf{0}$ for Weight-Pruning and $\hat{\boldsymbol{W}}$ for Bit-Pruning) is evaluated using Hoyer-Square.
>
> >3. All convolutional layers using Bit-pruning?
>
> Yes.
> We use Weight-Pruning and Bit-Pruning for all the layers (both $3\times 3$, $1\times 1$ convolution, and depth-wise convolution in ConvNeXt), including the input layer.
> We've clarified this point in the experimental section (Sec. 4.1).

---

> ### Author Response · Authors · 2022-11-19
> **Authors Response to Reviewer Rw8U (2/2)**
>
> >4. Why all layers use the same $\lambda$?
> Probably earlier layers would had benefit for less sparse weights retaining in that way some of the degradation that would otherwise propagate to the rest of the network.
> Another motivation for this is that early layers have comparatively much fewer OPs than deeper layers so the potential speedup might not be worth it.
>
> As a preliminary study investigating the effectiveness of Bit-Pruning, we use the same $\lambda$ for entire layers.
> But as $\textbf{Rw8U}$ points out,  we can use a different value of $\lambda$ for each layer to get a better tradeoff, considering the different impacts on the energy consumption of each layer.
> We have made the following changes to reflect this valuable feedback.
>
> We've changed $\lambda$ to $\lambda^{(l)}$ in the entire manuscript to emphasize that the $\lambda$  could be chosen layer-wise (Sec.3.2), clarified that we used same $\lambda$ for all layers in the experimental setup,  and added the following discussion about the reason for our choice (adopting same $\lambda$ for all layers) into Appendix A.5.1.
>
> The $\lambda$ can be different for each layer.
> It can be set proportional to the number of operations for the given layers, or it could take the susceptibility of the weight change to the output layer into account.
> Although it requires additional ingenuity to approximate the gradient w.r.t. $\lambda$ (because of the non-differentiability of $\arg\min$ operation), learning the layer-wise $\lambda$ end-to-end may also be possible.
> In any case, it will introduce an additional hyper-parameter to be tuned.
> In this study, we focus on comparing the Weight-Pruning and Bit-Pruning in a simple and fair setup as possible.
> Therefore we left the exploration in this direction for future work.
>
> Relating to this comment, we've conducted the new experiments using the difference value of $\lambda$ ($\lambda=0.1$, same for all layers).
> We've added the results and discussion in Appendix E.2.
>
> Please also refer to the related discussion from other reviews ($\textbf{MKH6}$-(2))

---

### Official Review · Reviewer_bvLy · 2022-10-28

**Confidence:** 5
**Clarity, Quality, Novelty And Reproducibility:** It is written clearly, and well organ…
**Correctness:** 3
**Technical Novelty And Significance:** 3
**Empirical Novelty And Significance:** 3
**Recommendation:** 6

**Strength And Weaknesses:**

Strength:

* The idea sounds interesting to me.
* The proposed bit-pruning method beats weight pruning across several experiments.
* This paper clearly describe the most related background work, ShiftAddNet, and explain why it is not direclty comparable.

Weakness:

* Some math formulation could be more concise and not necessarily complex. Or you can give more high-level insights before diving into the math. Also, the figure quality is not enough, I strongly encourage the authors to revise their figure to the best level you can draw.
* Apart form the energy cost, latency comparisons or number of operations are expected and should be more direct.
* More quantization works and bit pruning works should also be compared with, I believe there are a lot bit-pruning works, either compare in a table with checks or compare in terms of experiments or both.

**Summary Of The Paper:**

The idea of this work is to first represent DNNs with add-shift-add format instead of mult-add. By doing so, they observe that the first accumulation step contribute a lot to the energy consumptions. As such, they propose the bit-pruning for the first add in terms of the shiftadd perspective.

The key is not representing weights in add-shift-add. Rather, the key is to regularize and prune the number of adds.

I like the idea and am willing to take a further look at the implementation later on.

**Summary Of The Review:**

In a nutshell, I recommend this paper due to their novel insights for bit pruning in terms of shiftadd representation of DNNs.

BTW, I wonder how this would be efficient and fast on GPUs? I understand that may need customized CUDA implementations but am curious which stage are you currently in.

Also, more quantization works and bit pruning works should also be compared with. More baslines and better figures would further strenthen this paper.

As for the difference with ShiftAddNet, I buy the explaination that they are not direclty comparable. ShiftAddNet reparameterizes DNNs with bit-shifts and adds. This work "translate" multiplications with shift and adds and prune the first accumulation process.

They can even be merged as a hybrid model, like ShiftAddNAS [1] did.

[1] ShiftAddNAS: Hardware-Inspired Search for More Accurate and Efficient Neural Networks, ICML 2022

---

> ### Comment · Reviewer_MKH6 · 2022-11-16
> **Question to the authors**
>
> Hey, I'm writing under your review, as you raise an interesting question about the implementation on GPUs.
>
> I understand this can be implemented in CUDA as much as they implemented their simulator for the experiments. However, I believe that in order to realize those gains, they might need to use a different architecture altogether such as FPGAs. But this would be a great question for the authors, to clarify how a real implementation could look like or what is needed for the approach to show benefits in real world experiments.

---

> > ### Author Response · Authors · 2022-11-19
> > **Efficient Implementation**
> >
> > Thank you for the compliment about efficient computation.
> > Your understanding is completely correct.
> >
> > Our approach is designed for the H/W, which could utilize unstructured sparsity in weight (and activation).
> > Some prior work utilizes the GPU for efficient execution of the network having unstructured sparsity (such as SparseRT by [Ziheng]).
> > We are also interested in the utilization of the unstructured sparsity on GPUs that is widely available.
> > Although, we consider the unstructured sparsity can be more naturally utilized by optimized CPU implementation, FPGAs, or dedicated accelerators designed for sparse processing.
> >
> > For example, NeuralMagic (https://neuralmagic.com) provides a solution to utilize the unstructured sparsity both in weights and activation by using the highly engineered L1 and L2 cache mechanism of Intel-based CPUs.
> > Several companies, such as Intel (Loihi2) and GrAI Matter lab (GrAI VIP), have recently released a new type of accelerator by incorporating in-memory architecture supporting the unstructured sparsity by its hardware design.
> >
> > In the original manuscript, the above discussion was unclear, and we also failed to discuss the techniques for  utilizing the sparsity on general-purpose CPUs, which are even more widely used than GPUs.
> > We revised the manuscript so that the above discussion can be more precise.
> >
> > We are currently evaluating the Bit-Pruning on the CPUs by utilizing a similar technique as the NeuralMagic.
> > We are also working on the evaluation using the data-flow type processors based on the in-memory computing architecture.
> > The evaluation of wall clock time and the actual energy consumption using the CPU and the dedicated processors will be presented as a subsequent separate work.
> >
> > Please also refer to our comment about this topic ($\textbf{bvLy}$-(3))

---

> ### Author Response · Authors · 2022-11-19
> **Authors Response to Reviewer bvLy (1/2)**
>
> We thank the reviewer $\textbf{bvLy}$ for clearly pointing out strengths, and we also appreciate comments for improving the manuscript.
> In the following paragraph, we clarify the concerns raised by the review.
>
> >1. You can give more high-level insights before diving into the math.
> ... the figure quality is not enough, I strongly encourage the authors to revise their figure to the best level you can draw.
>
> We agree with $\textbf{bvLy}$ that a more high-level explanation before diving into the detailed mathematical formulation is helpful for readers.
> We've added the intuitive explanation at the beginning of Sec.3.
> In addition, we've improved the quality of  Fig.1 and Fig.2 so that this study's motivation and main contribution can be more intuitively understood.
>
>
> >2. Latency comparisons or number of operations are expected
>
> We agree with $\textbf{bvLy}$ regarding the operation count.
> This is a fundamental statistic to see when\/why  Bit-Pruning has benefited over Weight-Pruning.
> We've added the following explanation in the experimental section.
>
> The number of $\texttt{mult}$ (in $\texttt{mult-add}$ network) and $\texttt{add}$ (in $\texttt{add-shift-add}$ network) can be compared by multiplying $M$ (number of bits for weight) by the results of $\texttt{add-shift-add}$ for the plots shown in Fig.5 and Fig.6.
> (Because the energy consumption is roughly given by $E_{\operatorname{mult}} \approx M E_{\operatorname{mult}}$ from Tab.1  in the case of ASIC, and we use Tab.1 to plot the figures).
> As can be seen from the plots (by multiplying  $M$), we need more $\texttt{add}$ in $\texttt{add-shift-add}$ than $\texttt{mult}$ in $\texttt{mult-add}$ to achieve the same accuracy.
> However, their ratio is significantly less than $M$; therefore, we'll get the better energy-accuracy tradeoff.
> From a different viewpoint, when $\sigma_{\operatorname{add}}$ is smaller than  $\sigma_{\operatorname{mult}}$, then the Bit-Pruned is more efficient (Eq.8), and vise versa.
> We realize $\sigma_{\operatorname{add}}<\sigma_{\operatorname{mult}}$
> by the proposed fine-grained bit-level pruning.
>
> We have not yet compared the wall-clock latency in actual H/W.
> And we are not sure whether $\texttt{add-shift-add}$  network (trained with Bit-Pruning) is superior to the $\texttt{mult-add}$ network (trained with Weight-Pruning) in terms of the latency (which may depend on the hardware, and it could be faster depending on the sparsity and parallelization because $\texttt{add}$ is usually faster than that of $\texttt{mult}$, e.g., on Intel CPUs).
> The main benefit of using bit-pruned $\texttt{add-shift-add}$ formulation is *energy* and *area* saving.
>
>
> >3. How would this be efficient and fast on GPUs? ...  curious which stage are you currently in.
>
> Thanks to the comments, we noticed the original manuscript was unclear at this point, and a discussion about some important studies was missing.
> To incorporate the comment, we've added a discussion about the scope of this study and related literature (which we missed including in the original manuscript) in the Sec.1 and Sec.2 to make the following point to be clear.
>
> As $\textbf{MKH6}$ kindly commented, our approach is designed for the H/W, which could utilize unstructured sparsity in weight and activation.
> There exists some prior work utilizing the GPU for efficient execution of the network having unstructured sparsity (such as SparseRT by [Ziheng]).
> Although, we consider the unstructured sparsity can be more naturally utilized by optimized CPU implementation, FPGAs, or dedicated accelerators designed for sparse processing.
> For example, NeuralMagic (https://neuralmagic.com) provides a solution to utilize the unstructured sparsity both in weights and activation by using the highly engineered L1 and L2 cache mechanism of Intel-based CPUs.
> Several companies, such as Intel (Loihi2) and GrAI Matter lab (GrAI VIP), have recently released a new type of accelerator by incorporating in-memory architecture supporting the unstructured sparsity by its hardware design.
>
> We are currently evaluating the Bit-Pruning on the CPUs by utilizing a similar technique as the NeuralMagic.
> We are also working on the evaluation using the data-flow type processors based on the in-memory computing architecture.
> The evaluation of wall clock time and the actual energy consumption using the CPU and the dedicated processors is left for future work (Sec.6.1).
>
> Please also refer to the related discussion from other reviews ($\textbf{MKH6}$-(1), $\textbf{79uo}$-(7))

---

> ### Author Response · Authors · 2022-11-19
> **Authors Response to Reviewer bvLy (2/2)**
>
> >4. As for the difference with ShiftAddNet, I buy the explanation that they are not directly comparable.
> ...
> This work "translate" multiplications with shift and adds and prunes the first accumulation process.
> They can even be merged as a hybrid model like ShiftAddNAS did.
>
> We appreciate the positive comments on our discussion with ShiftAddNet.
> And we also thank $\textbf{bvLy}$ for letting us know about the ShiftAddNAS, which we did not know.
> We agree that our Bit-Pruning can be combined with their work; e.g., the best combination of $\texttt{shift-add}$  and our $\texttt{add-shift-add}$ are automatically searched.
> By combining the work, we may realize a more efficient  $\operatorname{mult}$ free network.
> We've added the discussion with ShiftAddNAS in the related works (Sec.5.)
>
>
> >5. More quantization works and Bit-Pruning works should also be compared with, ...,  either compare in a table with checks or compare in terms of experiments or both.
>
> We've clarified the discussion with the other method in Sec.3.4.
> We discuss the relationship with another study for low-energy DNN, such as non-uniform quantization, PoT quantization, and unstructured pruning,  in terms of their functionality.
> Among them, we consider the experimental comparison with unstructured pruning to be most meaningful because we can compare them based on similar computational architectures (please refer to the previous discussion), and ours can be considered as an extension of unstructured pruning in some sense.
>
> Thanks to this comment, we found the discussion could be highlighted more, and we rewrote the section to highlight their relations more clearly.
> Inspired by the suggestion for using *table with checks*, we highlighted the similarity and differences using highlighted text.

---

> ### Comment · Reviewer_bvLy · 2022-11-24
> **Response**
>
> I thank the authors for their feedback. After reading your rebuttal, I would like to maintain my score and recommend acceptance.

---

### Author Response · Authors · 2022-11-19
**Authors Response to All the Reviewers**

We thank all the reviewers for clearly pointing out strengths and weaknesses.
In response to the constructive suggestions provided by all four reviewers, we have spent the past week conducting new experiments and revising our manuscript to incorporate the new information.
By incorporating the suggestion, we are confident that the revised manuscripts have now much clearer.
We'll respond to each of the valuable comments in each review comment thread.

Along the course of incorporating valuable suggestions, we've conducted several new experiments.
And the results are added to the main paper and Appendix.
Due to the time limitation, we've conducted the new experiments only on CIFAR10 using 8-bit activation.
Note that the code we'll open source includes the script to run the new experiments on other settings (same as our main experiment result shown in Fig.4).

---

### Decision · Program_Chairs · 2023-01-20

**Decision:**

Accept: poster

**Justification For Why Not Higher Score:**

The specific strategy for network compression is not broadly ready for prime-time without some additional linear algebra library development.  It's an interesting idea, but the reviewers were not overwhelmingly positive that it is a priority for a spotlight.

**Justification For Why Not Lower Score:**

The topic of computational and energy efficiency in NN deployment is an important one.  This paper makes a strong argument for a specific view on basic compute operations that can lead to improved efficiency/accuracy tradeoff.

**Metareview: Summary, Strengths And Weaknesses:**

The submission proposes a low-energy model compression strategy based on an analysis of the basic operation of a dot product.  The strategy takes the perspective of a basic "add-shift-add" operation, and then looks at pruning the number of operations by sparsification.  The rebuttal/discussion process clarified the applicability and hardware assumptions of this model, which in certain circumstances was demonstrated to increase efficiency / decrease energy usage favorably to other methods.  The application requires (i) the appropriate kind of model compression, (ii) appropriately optimized hardware, and (iii) an appropriately optimized linear algebra library.  As such, the immediate application may be limited to specialized settings.  It is a valuable exercise to explore interactions between statistical accuracy, and computational and energy efficiency, and this submission does so in a concrete and interesting way.  The reviewers were generally appreciative of the contributions of the submission, with a consensus that it is at least above the threshold for acceptance.

**Note From Pc:**

if the above contains the word "oral" or "spotlight" please see: "oral" presentation means -> notable-top-5% and "spotlight" means -> notable-top-25%. As stated in our emails, we are disassociating presentation type from AC recommendations